# Impact of Electric Arcs and Pulsed Electric Fields on the Functional Properties of Beta-Lactoglobulin

**DOI:** 10.3390/foods11192992

**Published:** 2022-09-26

**Authors:** Rock-Seth Agoua, Laurent Bazinet, Eugène Vorobiev, Nabil Grimi, Sergey Mikhaylin

**Affiliations:** 1Dairy Research Center (STELA), Institute of Nutrition and Functional Foods (INAF), Université Laval, Québec, QC G1V 0A6, Canada; 2Laboratory of Food Sustainability (EcoFoodLab), Food Science Department, Université Laval, Québec, QC G1V 0A6, Canada; 3Laboratoire de Transformation Alimentaire et Procédés ÉlectroMembranaires (LTAPEM, Laboratory of Food Processing and Electromembrane Processes), Food Science Department, Universiteé Laval, Québec, QC G1V 0A6, Canada; 4Laboratory of Agro-Industrial Technologies, University of Technology of Compiègne, 60203 Compiègne, France

**Keywords:** β-lactoglobulin, functional properties, pulsed electric field, electric arc, proteins

## Abstract

Beta-lactoglobulin (β-lg) is a major whey protein with various techno-functional properties that can be improved by several treatments. Therefore, the objective of this study was to explore the impact of green high-voltage electrical treatments (HVETs)—namely, pulsed electric fields and electric arcs—on the functional properties of β-lg. Both emulsifying and foaming stability and capacity, as well as the hygroscopicity of non-treated and pretreated β-lg, were explored. The results demonstrated that the emulsifying capacity and stability of pretreated samples increased by 43% and 22% when compared to native β-lg, respectively. Likewise, the pretreated β-lg displayed better foaming stability compared to native β-lg. In addition, the HVETs significantly decreased the hygroscopicity of β-lg (by 48% on average), making it a good ingredient with reduced hygroscopicity for the food industry.

## 1. Introduction

Whey is a byproduct resulting from the production of cheese, Greek yogurt, and casein. Its proteins are considered highly nutritious and have been used as ingredients in various food products for their desirable functional properties, such as gelation, emulsification, foaming, flavor-binding properties, etc. [1,2,3]. However, the functional properties of whey protein could be improved via processing methods that modify protein structure [1,4,5]. Previous studies have provided considerable amounts of information on the molecular structural changes of proteins induced by various non-specific treatments, including chemical, enzymatic, and physical methods [4,6]. However, physical modification has been reported to be the easiest method to apply for the improvement of proteins’ functional properties. For example, many authors have reported that limited heat treatment can improve the foaming and emulsifying properties of β-lg [3,5,6,7,8]. Indeed, β-lg is the most abundant whey protein, with multiple functional properties that have been widely described in the literature over the past decades. The improvement of its emulsifying and foaming properties after heat treatment has been reported in several studies [7,9]. For example, Resch et al. [10] and Renard et al. [11] investigated the gelation properties of β-lg, while its foaming and emulsifying properties were studied by Moro et al. [7], Hu et al. [12], and Pein et al. [13]. Moreover, in recent years, numerous emerging technologies have been investigated to improve the functional properties of whey protein products, including high hydrostatic pressure, ultrasound, pulsed light, etc. For example, Ibanoglu [14] and Ali et al. [15] reported that high pressure improved some of the functional properties (i.e., foaming and emulsifying stability) of β-lg, whereas Pittia et al. [16] reported a decrease in some functional properties (i.e., foaming capacity and stability) of β-lg. The improvement of functional properties of β-lg induced by high-intensity ultrasound has been reported by Chandrapala et al. [17] and Shen et al. [1], whereas Fernandez et al. [18] reported the improvement of the foaming properties of β-lg, processed with pulsed light. However, these emerging technologies present some disadvantages (e.g., considerable energy consumption, relatively long treatment duration, etc.) when compared with high-voltage electrical treatment (HVET). Indeed, compared to some emerging technologies, HVET demonstrate low energy consumption and better improvement of various food processes [19,20]. Moreover, referring to the results reported by Pittia et al. [16], Arzeni et al. [21], Zhang et al. [22], and Sui et al. [23], it has emerged that HVET seem to be more beneficial compared to high hydrostatic pressure and high-intensity ultrasound, which have shown lower efficiency in improving whey proteins’ functional properties.

There are two types of HVET—namely, pulsed electric field (PEF), and high-voltage electrical discharge, also called “Electric Arc” (ARC) [19,24,25,26,27]. These two HVET modes are differentiated by the nature of the electrodes that constitute them, as well as by the various phenomena occurring during the treatments, as has been reported by many authors [19,28,29]. Indeed, as reported by Boussetta et al. [28], electric arcs are formed when the thin ionized vapor channels (“streamers”) initiated at the tip of the point electrode reach the ground-plane electrode, inducing the production of chemically active species such as hydrogen peroxide, ozone, and free radicals (OH^•^, H^•^, O^•^, etc.), along with physical phenomena such as UV light, shockwaves, and cavitation bubbles. However, in PEF mode, the solution is placed between two plane electrodes, and only the generation of chemically active species occurs during this treatment. The abovementioned phenomena occurring during ARC and PEF modes have demonstrated an impact on β-lg’s structure and hydrolysis performance [26,27]. It is worth noting that PEFs are already successfully applied in the food industry, while ARC mode is still being explored in the laboratory and in pilot studies. Regarding the influence of HVET on the functional properties (foaming, emulsification, etc.) of food proteins, only PEF mode has been studied so far, while ARC mode remains unexplored. For instance, Sui et al. [23] studied the effect of a PEF on the functional properties of whey protein isolate, Sharma et al. [30] nvestigated the effect of a PEF on the functional properties of bovine milk, and Odriozola-Serrano et al. [31] and Zhang et al. [22] studied the effects of PEFs on the quality and functional properties of canola proteins. Indeed, except for Sui et al. [23], who reported that the PEF did not affect the physicochemical properties or the emulsifying properties of whey protein isolates, the abovementioned authors observed significant improvement in the functional properties of bovine milk proteins. The effects of PEFs on the functional properties of other food proteins have also been reported by many authors. For example, Zhang et al. [22] reported the improvement of the structural and functional properties (i.e., solubility, water-holding capacity, oil-holding capacity, emulsifying capacity and stability, foaming capacity, and foam stability) of canola protein pretreated with a PEF, whereas Li et al. [32] reported significant changes in solubility as wells as in the surface free sulfhydryls and hydrophobicity of soybean protein isolates under the different conditions of PEF. Manzoor et al. [33] and Xiang et al. [34] have reported the improvement of the rheology, color, and physicochemical properties of PEF-treated almond and soy milks, respectively. Nevertheless, the effects of HVETs on the functional properties of pure β-lg protein have never been studied. Indeed, few studies have been dedicated to the effects of HVET on the functional properties of pure food proteins. To the best of our knowledge, only Perez and Pilosof [35] have investigated the effects of PEFs on the gelation properties of β-lg. However, the effect of the ARC mode of HVET on the functional properties of food proteins in general, and β-lg in particular, has not yet been the subject of any publication, making this study very innovative. Hence, the main objective of this study was to investigate whether emergent HVET can improve the foaming, emulsifying, and hygroscopic properties of β-lg, and to compare them with conventional heat-based pretreatments. These functional properties were chosen due to their great interest to the food industry, aiming at the formulation of a wide range of food products [36,37,38,39].

## 2. Materials and Methods

Bovine β-lactoglobulin with a purity of 98% was obtained from Davisco Foods International, Inc. (Eden Prairie, MN, USA), commercial vegetable oil was purchased from the Metro supermarket (Quebec, QC, Canada), and Na_2_SO_4_ (No.-CAS: 7757-82-6) was purchased from Sigma-Aldrich Canada Co. (Oakville, ON, Canada)

### 2.1. Configuration of the HVET System

The system used for high-voltage electrical treatments (HVETs) for β-lg pretreatments was the same as previously described by Agoua et al. [26,27], consisting of 1 L cylindrical treatment chambers distinguished by the nature of the electrodes that constitute them. The PEF treatment chamber was equipped with two plate electrodes (diameter = 35 mm), whereas the ARC chamber consisted of a needle electrode (diameter = 10 mm) and one plate electrode (Figure 1). The HVET treatment chambers were connected to a high-voltage pulse generator (Tomsk Polytechnic University, Tomsk, Russia) that can provide a maximal voltage and peak power of 40 kV and 4 × 10^5^ kW, respectively. A positive pulse voltage was applied to the upper electrode at a frequency of 1 Hz. For the PEF treatment system, the distance (*d*) between the electrodes was adjusted to 20 mm, and to 5 mm for the ARC system. The pretreatment conditions were set as described by Mikhaylin et al. [25] and in the few research works in which PEF was used to improve the functional properties of food proteins [22,23,25].

### 2.2. Protocol of Protein Pretreatments

The β-lg electrical pretreatments were performed as described by Mikhaylin et al. [25], with some modifications. Briefly, the treatment chambers were filled with 300 mL of pure β-lg solution (20% (*w*/*v*)), which was prepared at 4 °C using distilled water 16 h before the treatments with the dried protein powder. This concentration was chosen from the preliminary tests as the highest possible for treating the protein solution in available HVET cells. The protein solution was treated with both HVETs for 1 and 10 min, corresponding to 30 and 300 pulses, respectively. The generator provided energy of 160 J per pulse, and the duration of each pulse was 10 µs. The conductivity (1765 μS/cm) and the pH (6.8) of the β-lg solutions were not significantly affected by HVET. It is noteworthy that HVETs are considered to be non-thermal treatments. Thus, no substantial increase in temperature during the pretreatments was observed, except for the PEF 10 min treatment, where the final temperature was around 50 °C due to the Joule heating.

The conventional thermal pretreatment of β-lg, which represents the positive control, was performed in order to compare its performance with that of the emergent HVET pretreatments. Briefly, 100 mL of β-lg solution (20% (*w*/*v*) was heated up to 85 °C in a water bath with temperature control and held for 5 min according to the literature [40,41,42,43,44]. All of the pretreatments were performed in triplicate, and the β-lg solutions (pretreated and non-treated) were freeze-dried and then stored in the freezer in hermetically sealed tubes until further analysis.

The various analyses of functional properties—namely, emulsifying, foaming, and hygroscopicity properties—were carried out using native and pretreated proteins. The native protein means non-pretreated protein within the framework of this study, thus representing the negative control.

### 2.3. Foaming Properties of β-lg

Foaming capacity and foam stability were measured according to the methods described by Mohanty et al. [45] and Lin [46], with some modifications. Solutions of 1 g of native and pretreated β-lg in 100 mL of distilled water were prepared. The initial volume of protein solutions (100 mL) was measured in a graduated cylinder. The solutions were transferred to a container and whipped for 5 min with an Oster 2599-033 commercial food mixer (Sunbeam Products Inc., Brampton, Ontario, Canada) at maximum speed (800 rpm). The whipped solution was immediately transferred into a graduated cylinder and the foam volume was measured. The foaming capacity (FC, in %) was calculated using Equation (1):(1)$$\mathrm{FC}=\frac{V_{foam}}{V_{i}}\;\times \;100$$
where *V**_foam_* (mL) is the foam volume, and *V_i_
*(mL) is the initial volume of the protein solution.

The foam stability was determined by measuring the variation of the foam volume after 0, 1, 5, 10, 30, 60, 120, and 180 min. The analyses were performed in triplicate.

### 2.4. Emulsifying Capacity of β-lg

The emulsifying capacity (EC) of β-lg was measured according to the methods described by Mohanty et al. [45], Moro et al. [7], and Shen et al. [1], with slight modifications. Briefly, 100 mL samples of native and pretreated β-lg solutions (pH = 6.8) were prepared in distilled water. Two milliliters of the protein solution were transferred to a 50 mL beaker. Commercial vegetable oil was added gradually (about 5 mL/min) during homogenization with an Ultra Turrax T25 mixer (IKA Works Inc., Wilmington, CN, USA) at a speed of 17,500 rpm, until the sudden drop in viscosity associated with the inversion of the emulsion was recorded.

The emulsifying capacity was expressed as the amount of oil per 100 mg of protein [46]. All experiments were performed in triplicate.

### 2.5. Emulsion Stability of β-lg

The emulsion stability (ES, in %) of β-lg was determined according to a modified version of the methods of Stone et al. [47] and Stone and Nickerson [48], using Equation (2). As previously described, 100 mL samples of native and pretreated β-lg solutions (pH = 6.8) were prepared in distilled water, and 5.4 mL of the dispersion was added to a 50 mL beaker with 10 mL of the commercial vegetable oil and homogenized with an Ultra Turrax T25 mixer (IKA Works Inc., Wilmington, CN, USA) at a speed of 9500 rpm for 2 min. The emulsions were transferred to 15 mL centrifuge tubes. The volume of the aqueous phase was reported after 24 h. The analysis was performed in triplicate.
(2)$$ES=\frac{V_{b}-V_{a}}{V_{b}}\;\times \;100$$
where *V_b_* (mL) is the volume of the aqueous phase before emulsification, and *V_a_* (mL) is the volume of the aqueous phase after 24 h.

### 2.6. Hygroscopicity of β-lg

The hygroscopicity of a β-lg powder was characterized by its final moisture content after being introduced to air at controlled relative humidity, under defined conditions as described by Ma et al. [49], with some modifications. About 1 g of β-lg samples was placed in Petri dishes kept in a desiccator filled with saturated Na_2_SO_4_ solution for one week, at room temperature. The powders were weighed after 24 h, and the difference in weight shows the hygroscopicity expressed in g of water absorbed per 100 g of dry β-lg powder. This experiment was performed in triplicate.

#### Statistics

One-way analysis of variance (ANOVA) with Tukey’s test was performed to identify mean differences in the values of the determined functional property parameters. Statistical analysis was performed using SigmaPlot software version 12.0 build 12.2.0.45 (Systat Software, Inc., wpcubed GmbH, Erkrath, Germany).

## 3. Results and Discussion

### 3.1. Foaming Properties of β-lg

#### 3.1.1. Foaming Capacity of β-lg

The foaming capacity of native as well as electrically and thermally pretreated β-lg samples is shown in Figure 2. When comparing the native β-lg with the pretreated β-lg, no statistically significant differences in foaming capacity were observed. When the foaming capacity of native β-lg was compared with that of the HVET-pretreated samples, only the β-lg pretreated by ARC for 10 min showed a significant difference (*p* < 0.05), exhibiting the highest foaming capacity (586.7%). A similar observation was made when comparing preheated β-lg with HVET-pretreated β-lg. Indeed, the foaming capacity of all pretreated samples was roughly similar, except for the 10 min ARC sample. However, when comparing the HVET mode (PEF vs. ARC), there was no clear evidence as to which HVET mode was the most efficient in improving the foaming capacity of β-lg. Nevertheless, concerning the HVET durations, 10 min pretreated samples seemed to have a better impact on β-lg’s foaming capacity than the 1 min pretreated samples. Indeed, for the ARC mode, there was a significant difference in foaming capacity when comparing 1 min and 10 min treatment durations. As for PEF, even though there was no significant difference between the 10 min and 1 min pretreated samples, we noted a slight improvement of the foaming capacity of the 10 min PEF sample (Figure 2).

The obtained results are consistent with those of Zhang et al. [22], who reported a significant improvement in the foaming properties of canola proteins pretreated with PEF. It is important to emphasize that the treatment parameters used by Zhang et al. [22] were similar to those used in this study. Moreover, the results are consistent with those of Fernandez et al. [18] as well as Siddique et al. [50], who indicated that the foaming capacity of β-lg and whey protein isolate, respectively, was considerably improved after pretreatment with pulsed light. Indeed, the first authors reported an increase of 12% in the foaming capacity of the pretreated β-lg solution (10 g/L) compared to the native sample. This is consistent with the results of the present study, where an increase of 11% was observed for β-lg pretreated with ARC for 10 min. In addition, Fernandez et al. [18] indicated that the foaming capacity of β-lg was significantly improved for lower concentrations of β-lg (e.g., 0.5–1.5 mg/mL), while more concentrated protein solutions (5–10 mg/mL) did not show significant differences between native and processed β-lg. This could explain the results obtained, since the concentrations of β-lg samples (native and pretreated) used in this study were 10 mg/mL. In contrast, Ibanoglu [14] and Pittia et al. [16] observed a decrease in the foaming properties of β-lg pretreated with high hydrostatic pressures. Additionally, Moro et al. [8] reported a significant improvement in the foaming capacity of preheated β-lg, while Shen et al. [1] reported in their study that the use of ultrasound significantly increased the foaming ability of whey proteins. Overall, the improvement in the foaming properties of β-lg may be essentially due to the impact of HVET on the structure of the protein. Indeed, as previously reported by Agoua et al. [26], HVET induces structural modifications in β-lg molecules, leading to a decrease in the numbers of β-sheets and random coils, as well as an increase in the number of α-helices. A similar hypothesis was proposed by several authors who attributed the improvement in the functional properties of the studied proteins to structural modifications induced by the abovementioned pretreatments [8,14,16,22]. However, the foaming properties of proteins are not only determined by their capacity to form foam. Thus, the breakdown time of the foam (foam stability) was also investigated.

#### 3.1.2. Foam Stability of β-lg

The impact of HVET on β-lg’s foam stability is shown in Figure 3. The foam stability is characterized by the variation in the produced foam volume as a function of time. Since the pretreated samples showed different behavior in terms of foam stability, the results are then discussed as a function of time. Thus, at the first 30 min after whipping, no significant differences were noted between the β-lg samples in terms of the stability of the foam. However, it was rather evident after 60 min that there were significant differences between the various β-lg samples. Indeed, when comparing native and preheated β-lg, no significant difference in foam stability was observed. Nonetheless, the foam stability was significantly different (*p* < 0.05) when comparing both native and preheated β-lg with HVET-pretreated samples. Indeed, the HVET-pretreated samples had 24% higher foam stability. Moreover, when comparing the HVET modes, it emerged that ARC-pretreated β-lg showed higher foam stability than PEF-preheated β-lg at 60 min. Regarding the HVET durations, the foam stability was quite similar for 1 min and 10 min of PEF, whereas a slight improvement was observed for 10 min of ARC as compared to 1 min of ARC.

Similar trends were observed after 90 min, where the foam stability considerably decreased, indicating a gradual disappearance of the foam. The foam stability was slightly higher for the preheated β-lg, but not significantly different as compared with the native sample. However, when comparing native β-lg with all HVET-pretreated samples, the foam was 63% more stable, showing the highest value for the ARC 10 min sample. After 120 min, the foam stability was close to 0% for native β-lg, while its values were still over 10% for the pretreated samples. The level of remaining foam was somewhat lower for the preheated β-lg compared to that of the 10 min ARC sample. Note that the level of foam remaining after 120 min in the 1 min AR and the, 1 and 10 min PEF samples was not significantly different to that of the preheated β-lg. It should be noted that the ARC-pretreated samples showed better foam stability than the PEF-pretreated ones. In addition, it can be seen that the complete disappearance of the foam occurred after approximately 120 min for native β-lg, while for the pretreated samples (preheated and HVET) the foam disappeared after more than 180 min. Thus, one can deduce from the above results that the pretreatments improved the foam stability of β-lg. Furthermore, the HVET-pretreated samples (mainly the ARC ones) showed better improvement in foam stability as compared to the native and preheated β-lg. Similarly, these results are consistent with the findings of Sun et al. [2], who observed an improvement in the foaming properties of reconstituted milk protein concentrates after high-intensity ultrasound treatment. Likewise, Dissanayake and Vasiljevic [51] reported the improvement of the foaming properties of whey proteins subjected to heat and hydrodynamic high-pressure treatments.

Finally, the impact of the β-lg pretreatment on the foam stability was different compared to the foaming capacity. Indeed, the foaming capacity of native β-lg was similar to that of the pretreated samples, except for the ARC 10 min sample. Conversely, the foam stability showed significant differences between native β-lg and all of the pretreated samples. This improvement in the foaming properties could be explained by the impact of pretreatments on the structure of the β-lg. Indeed, the formation of foam is favored by the exposure of the hydrophobic groups of the protein, as reported by Siddique et al. [50]. According to Shen et al. [1], hydrophobic interactions are of major importance for the stability, conformation, and functional properties of proteins. Moreover, the foam formation depends on many other factors, such as the physical and chemical properties of the proteins. Likewise, the disulfide bonds that stabilize the secondary and tertiary structures of the protein may also influence its functional properties. Thus, one can speculate that the different phenomena taking place during HVET treatments (e.g., chemically active species for PEF, shockwaves, cavitation bubbles, infrared, ultraviolet lights, etc., for ARC) induce conformational changes in the β-lg molecules, leading to their transient state, as demonstrated by Agoua et al. [26]. Thus, the pretreatments led to the exposure of hydrophobic regions initially buried in the internal structure of the native protein. Furthermore, as reported by Singh et al. [52], the applied external electric field could affect the polarization and the distribution of the electron density of the dipole moments of some bonds in the polypeptide chains. Thus, the higher values of foam capacity and stability exhibited by HVET-pretreated β-lg in this work could be attributed to its partial denaturation. This finding is consistent with several previous studies dealing with the processing of proteins to improve their foaming properties [8,12,14,17,18].

### 3.2. Emulsifying Properties of β-lg

#### 3.2.1. Emulsifying Capacity of β-lg

The emulsifying capacity of pretreated and native β-lg samples is shown in Figure 4. An analysis of variance demonstrated significant differences (*p* < 0.05) between the native and pretreated samples. Indeed, the emulsifying capacity value was significantly higher for native β-lg as compared with the preheated β-lg, indicating that 22% more oil was needed for the inversion of emulsion. This suggests that preheating of β-lg improved its emulsifying capacity. A similar observation was made when comparing native β-lg with HVET-pretreated samples—less oil (24%) was needed to reach emulsion inversion for all HVET samples. However, there were no significant differences in emulsifying capacity when comparing the preheated samples with the HVET ones. Furthermore, when comparing the HVET-pretreated samples with one another, the emulsifying capacities were roughly similar.

#### 3.2.2. Emulsion Stability of β-lg

Figure 5 shows the emulsion stability of native and pretreated β-lg samples. One can observe that all of the pretreated samples (preheated and HVET) had higher emulsion stability values than the native sample. In fact, when comparing native β-lg with the preheated sample, there was a 30% improvement in emulsion stability. Likewise, there was significant difference in emulsion stability when comparing native β-lg with the HVET-pretreated samples. Nonetheless, when the preheated β-lg was compared with the HVET-pretreated samples, there were no significant differences in the emulsion stability. Furthermore, when comparing the HVET modes, the PEF mode showed better emulsion stability than the ARC mode—especially the 10 min PEF one. With regards to the HVET durations, the 10 min pretreatments seemed to improve the emulsion stability further than the 1 min treatments. Interestingly, at least one HVET mode showed overall better improvement in emulsifying and foaming properties than the preheated sample. These results are consistent with those of Li et al. [32], who reported an improvement in the emulsifying properties of soybean protein isolates pretreated with PEFs. Likewise, Shen et al. [1] also observed an increase in the emulsifying properties of whey protein treated with high-intensity ultrasound. However, on the ionic strength, as reported by Kim [53] and Moro et al. [7] observed a decrease in the emulsifying properties of β-lg subjected to heat treatments. Other authors have also demonstrated that large fractal aggregates of β-lg formed by heating were not able to improve foam stability [8]. However, in this study, all pretreatments, including heat treatment, showed overall better improvement of both foaming and emulsifying properties than the native β-lg. Such a difference could be due to the concentration of β-lg used as well as the performed treatment durations. Obviously, the experimental conditions used in this study were different from those in the aforementioned studies. In addition, our results are consistent with those reported by Sun et al. [2], and may be explained by the more favorable orientation of proteins resulting from the structural changes induced by the pretreatments and the integration of oil bubbles in the emulsion. Indeed, the abovementioned authors noted improvements in the emulsifying properties of milk protein concentrates, attributable to the effects of ultrasound treatments. According to the same authors, cavitation phenomena occurring during ultrasonication can emulsify liquids in seconds, whether in a batch or a continuous system [2]. Similarly, as previously mentioned, the different phenomena occurring during HVET treatments can individually or synergistically affect the structure of the protein. Indeed, as mentioned above, referring to the study of Agoua et al. [26], both PEF and ARC pretreatments induced conformational changes in β-lg. Additionally, according to Moro et al. [7]*,* foaming and emulsifying properties are closely related to the structural changes undergone by the protein. Furthermore, emulsions are thermodynamically unstable systems, and can be destabilized by various mechanisms, including phase coalescence of dispersed droplets, flocculation, inversion, and aggregation [2,54]. Indeed, as reported by some authors, the instability of emulsions may occur in the presence of a surfactant that is not sufficient to cover the entire interface created during homogenization [2,55]. Likewise, the factors affecting the foam stability (e.g., hydrophobic interactions) are similar to those affecting the stability of the emulsion. Therefore, as expected, the pretreatments also led to an increase in emulsifying properties, due to the exposure of more hydrophobic groups resulting from partial unfolding of the protein molecules. Indeed, surface hydrophobicity is an important factor that generally improves the emulsifying properties, since a correlation between the emulsifying properties and the surface hydrophobicity has been established for a number of proteins—both native and denatured [2,7]. In addition, the increase in surface hydrophobicity also depends on the charge distribution within the protein molecule and the ionic strength, as reported by Kim et al. [56]. Moreover, as previously mentioned, the dipole moments and distribution of the electron density of some bonds in polypeptide chains of the protein molecules may be affected by the applied external electric field [49].

### 3.3. Hygroscopicity of β-lg

As shown in Figure 6, the hygroscopicity of β-lg powder was affected by the pretreatments. Indeed, when native β-lg was compared with the preheated sample, there was no significant difference in the hygroscopicity, although a slight decrease could be noted for the preheated sample. However, when comparing both native and preheated β-lg with HVET-pretreated samples, the hygroscopicity decreased by 48% on average. Although Figure 6 appears to show differences between the HVET samples, the statistical analysis indicated that there were no significant differences in the hygroscopicity when comparing the HVET modes. The only significant difference observed was between the 1 min and 10 min PEF treatments. It is therefore difficult to clearly confirm which mode better improves the hygroscopicity of β-lg. Regarding the HVET treatment durations, the hygroscopicity of the 1 min HVET pretreatments was lower than that of the 10 min ones, with 1 min PEF having the lowest hygroscopicity value. These low hygroscopicity values obtained for the HVET-pretreated samples could be due to the structural modifications induced by the different phenomena occurring during the HVET pretreatments. In fact, hygroscopicity is closely linked with the physicochemical properties of the protein—mainly particle size, since moisture uptake occurs principally on the particle surface [57,58,59]. Moreover, according to O’Donoghue et al. [58], powder hygroscopicity increases linearly with decreasing particle size. As reported by Dissanayake and Vasiljevic [51], high hydrodynamic pressure induces conformational rearrangement of whey proteins by consequently reducing the stability of the hydrophobic core. Then, the aggregation of protein molecules via exposed hydrophobic groups may result in larger particles. Therefore, the possible aggregation induced by HVET would have increased the particle size, resulting in a decrease in hygroscopicity.

The above results suggest that the HVET-pretreated β-lg powders (especially the 1 min pretreated ones) could undergo fewer changes during storage and, could be more stable in food formulations—for example, in improved flours intended for confectionery or baking. Indeed, according to some authors, powders with lower hygroscopicity show better stability and have great potential to be used as food ingredients [49,60,61]. Additionally, as reported by many authors, a powder with a hygroscopicity value less than or equal to 10% is considered to be a good non-hygroscopic powder [62,63]. From this perspective, HVET-pretreated powders seem to be more advantageous compared to native and heat-pretreated ones.

## 4. Conclusions

The present study showed that HVET pretreatments can significantly improve the multiple functional properties of β-lg. Both foaming capacity and stability were improved after HVET with ARC for 10 min, showing an 11% increase in foaming capacity. The studied pretreatments led to the improvement of both the emulsifying capacity and stability of β-lg by 43% and 22%, respectively. Furthermore, the lowest hygroscopicity was observed for the HVET-pretreated β-lg samples. Indeed, HVET decreased the hygroscopicity of β-lg by 48%, making these samples good low-hygroscopicity ingredients that could be used in multiple food formulations. Interestingly, ARC pretreatments, whose impact on the functional properties of food proteins was investigated for the first time, showed better results in terms of improving the majority of the studied functional properties of β-lg. This represents an innovative scientific contribution, offering interesting perspectives to efficiently improve the functional properties of food proteins. Moreover, it emerges from the present study that HVET seems to be more efficient than conventional heating in terms of improving the functional properties of β-lg.

It would be interesting to study the effects of HVET on the functional properties of complex food protein systems (e.g., whey, soy, or pea protein isolates). Other potential perspectives could include the study of the impact of HVET (especially ARC mode) on other functional properties, such as solubility, water retention, gelling, etc. Finally, more in-depth studies of the HVET-pretreated protein structure using—for example—NMR and X-ray crystallography would be interesting to carry out in order to better understand their impact on the functionality of proteins.

## Figures and Tables

**Figure 1 foods-11-02992-f001:**
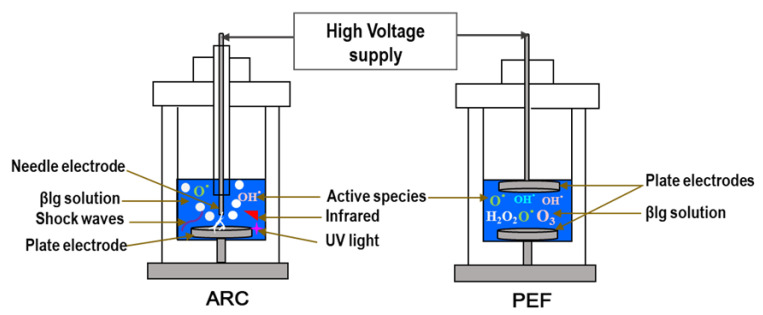
High-voltage electrical treatment system consisting of pulsed electric field (PEF) and electrical arc treatment chambers used for the pretreatment of β-lactoglobulin (β-lg) solutions.

**Figure 2 foods-11-02992-f002:**
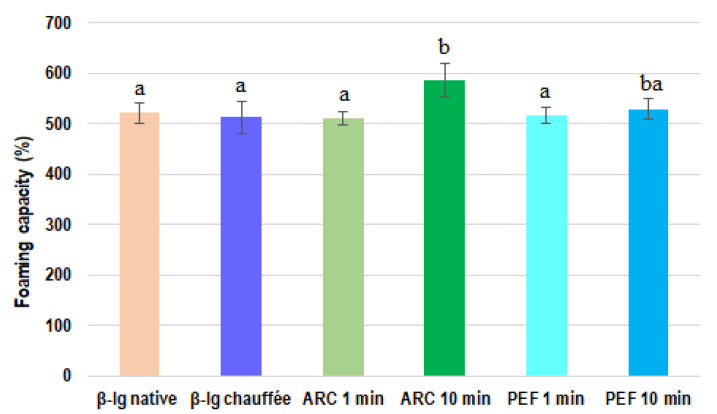
Foaming capacity (%) of native, preheated, and HVET-pretreated β-lg samples at pH 6.8. Different letters indicate statistically significant differences (*p* < 0.05). Treatments with at least one letter in common are not significantly different. Error bars represent standard deviations.

**Figure 3 foods-11-02992-f003:**
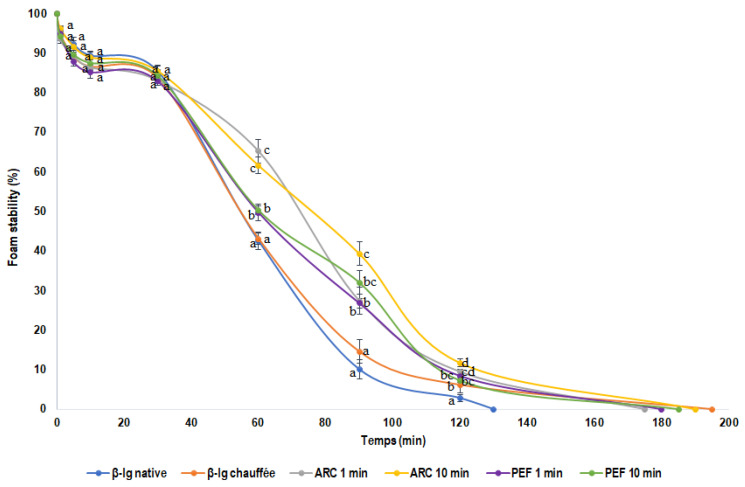
Foam stability of native, preheated, and HVET-pretreated β-lg samples at pH 6.8. Different letters indicate statistically significant differences (*p* < 0.05). Treatments with at least one letter in common are not significantly different. Error bars represent standard deviations.

**Figure 4 foods-11-02992-f004:**
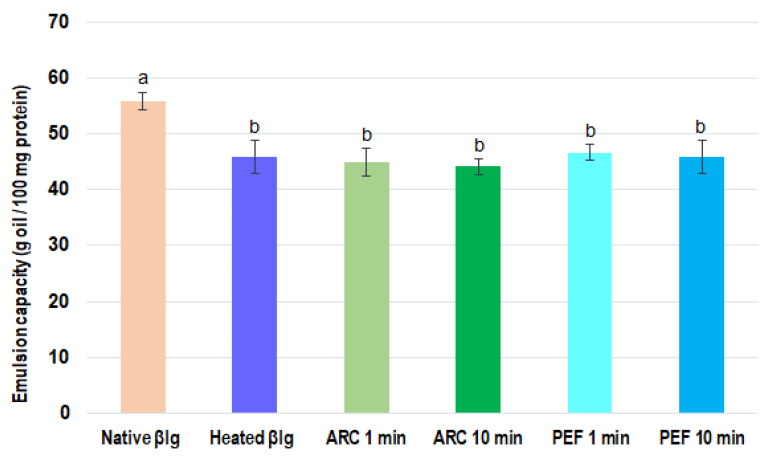
Emulsifying capacity of native, preheated, and HVET-pretreated β-lg samples at pH 6.8. Means with different letters indicate statistically significant differences (*p* < 0.05). Treatments with at least one letter in common are not significantly different. Error bars represent standard deviations. The colors are to better differentiate various β-lg treatments.

**Figure 5 foods-11-02992-f005:**
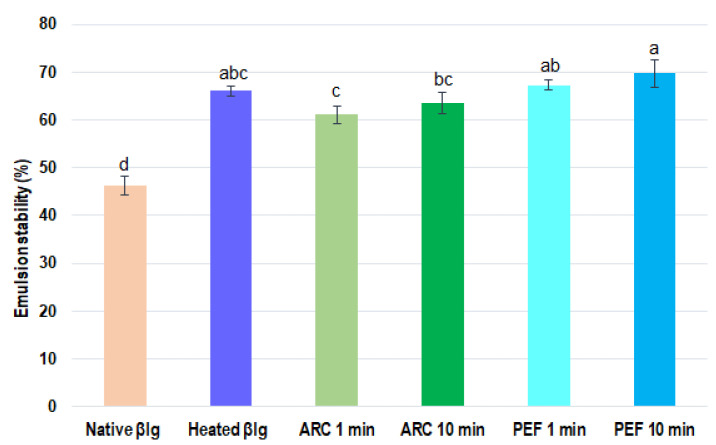
Emulsifying stability of native and pretreated (HVET and preheated) β-lg samples. Means with different letters indicate statistically significant differences (*p* < 0.05). Treatments with at least one letter in common are not significantly different. Error bars represent standard deviations. The colors are to better differentiate various β-lg treatments.

**Figure 6 foods-11-02992-f006:**
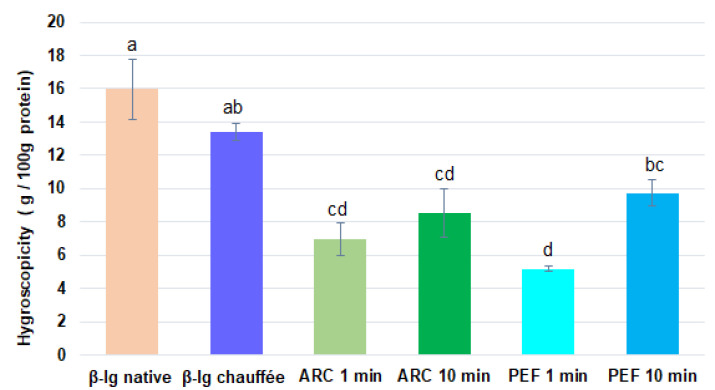
Hygroscopicity of native and pretreated (HVET and preheated) β-lg samples. Means with different letters indicate statistically significant differences (*p* < 0.05). Treatments with at least one letter in common are not significantly different. Error bars represent standard deviations. The colors are to better differentiate various β-lg treatments.

## Data Availability

The data used to support the findings of this study can be made available by the corresponding author upon request.

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
