# Peer review of "Impact of Electric Arcs and Pulsed Electric Fields on the Functional Properties of Beta-Lactoglobulin"

_foods, 2022, doi:10.3390/foods11192992_

Round 1

Reviewer 1 Report

Line 381. Change the word sonication to ultrasonication. Remember that the treatments are carried out in the acoustic spectrum of ultrasound.

Author Response

Thank you for this comment. The correction was made as requested.

Reviewer 2 Report

This research article reports observations on the effects of high voltage electrical treatments (HVET), namely microsecond pulsed electric field (PEF) and electric arc (ARC) exposures, on functional food properties of beta-lactoglobulin (b-lg). Several similar studies on the effects of PEF on other proteins have been published and are reviewed in the Introduction. However, this is stated to be the first report on the effects of ARC exposures on b-lg. HVET pretreatment effects were compared to those of thermal pretreatment and no treatment. Functional properties including foaming, emulsification, and hygroscopicity are measured using modifications of published methods, and analyses are performed in triplicate. The important conclusions are that HVET (especially ARC) pretreatments improved foaming and emulsifying capacities and stabilities and decreased hygroscopicity compared to thermal pretreatment and no treatment. This overall informative, observational article on the effects of HVET processing on functional food properties is appropriate for publication in Foods.

Minor revisions are recommended in order to address the following comments.

- The Introduction section could use a brief paragraph with more background on the ARC mode, especially with regard to the various physical phenomena that occur during ARC, such as shock waves. This is mentioned in a phrase in the Discussion section, but it is an important distinction to state.

- Was the value of temperature used in thermal pretreatments (85 degC) predicted to occur during HVET, e.g., from Joule heating? If so, please state this. If not, please address whether Joule heating is expected and what temperatures are predicted to occur during 1 min versus 10 min HVET treatments.

- Fix small formatting errors of different types throughout the manuscript, such as:

            - On line 112, is the power output from the pulse generator 4 x 10^5 kW?

            - Section and subsection headings are indented too far.

            - Abbreviate all units, including time, e.g., ‘minutes’ in lines 262-275.

            - Consistently use numerals for values, e.g., ’10 min’ instead of ‘ten minutes’ on line 361.

- Confirm that liquid solutions for ARC and PEF exposures consist only of beta-lactoglobulin in distilled water. If not, what is the composition?

- Show for comparison two representative oscilloscope captures of potential (and current if available) waveforms for ARC and for PEF treatments used here. 

 - State within figure captions what the error bars signify, e.g., one standard deviation, and how symbols such as ‘ba’ and ‘bc’ indicate a significant difference, e.g., a difference between ‘a’ and ‘b’? Or is ‘a’ different from ‘b’? Defining letters or symbols is necessary for clarity.

- Provide values for the electrical conductivity and pH measurements before and after treatments.

- Possible molecular-level changes induced by HVET treatments that could cause the reported results are discussed. Additionally, some techniques and experiments to verify these molecular changes should be mentioned as recommendations for future work. 

- Minor formatting corrections are needed in References, such as:

            - Title capitalization in [8] and [36],

            - Title abbreviation in [42]

- Conflicts of Interest and Data Availability Statements are needed.

Author Response

Comment 1:The Introduction section could use a brief paragraph with more background on the ARC mode, especially with regard to the various physical phenomena that occur during ARC, such as shock waves. This is mentioned in a phrase in the Discussion section, but it is an important distinction to state.”

Answer 1: Thank you for this relevant comment. A small paragraph was added as you suggested (see lines 65-75 as highlighted in the manuscript)

Comment 2:Was the value of temperature used in thermal pretreatments (85 degC) predicted to occur during HVET, e.g., from Joule heating? If so, please state this. If not, please address whether Joule heating is expected and what temperatures are predicted to occur during 1 min versus 10 min HVET treatments.

Answer 2: Thank you for raising this point. Indeed, HVET (both PEF and ARC) are considered as non-thermal treatments. We did not observe an increase in temperature during the pretreatments except for the PEF 10 min one when an increase in temperature up to 50°C was observed. However, this temperature is still low compared to that of the conventional heat treatment (see lines 138-142 as highlighted in the manuscript).

Comment 3:Fix small formatting errors of different types throughout the manuscript, such as:

            - On line 112, is the power output from the pulse generator 4 x 10^5 kW?

            - Section and subsection headings are indented too far.

            - Abbreviate all units, including time, e.g., ‘minutes’ in lines 262-275.

            - Consistently use numerals for values, e.g., ’10 min’ instead of ‘ten minutes’ on line 361.

Answer 3: Fixed as requested

Comment 4: Confirm that liquid solutions for ARC and PEF exposures consist only of beta-lactoglobulin in distilled water. If not, what is the composition?

Answer 4: Dear reviewer, yes, the pretreatments were performed with a pure (98%) beta-lactoglobulin solution (β-lg dissolved in distilled water) as mentioned in the protocol (see L.105).

Comment 5: Show for comparison two representative oscilloscope captures of potential (and current if available) waveforms for ARC and for PEF treatments used here.” 

Answer 5: This study and the entire research project was not intended to study HVET waveforms. Therefore, we did not focus on that and the captures are not available

Comment 6:State within figure captions what the error bars signify, e.g., one standard deviation, and how symbols such as ‘ba’ and ‘bc’ indicate a significant difference, e.g., a difference between ‘a’ and ‘b’? Or is ‘a’ different from ‘b’? Defining letters or symbols is necessary for clarity.

Answer 6: The captions were revised as requested.

Comment 7: Provide values for the electrical conductivity and pH measurements before and after treatments.”

Answer 7: We have provided both pH and conductivity values and reported that they were not significantly affected by the pretreatments (see L.138-142).

Comment 8: Possible molecular-level changes induced by HVET treatments that could cause the reported results are discussed. Additionally, some techniques and experiments to verify these molecular changes should be mentioned as recommendations for future work.”

Answer 8: The sentence was added as suggested (see L. 487-490)

Comment 9: Minor formatting corrections are needed in References, such as:

            - Title capitalization in [8] and [36],

            - Title abbreviation in [42].”

Answer 9: Revised as requested.

Comment 10: Conflicts of Interest and Data Availability Statements are needed.”

Answer 10: completed as requested.

Reviewer 3 Report

Dear Editor,

thank you for inviting me to review this manuscript. The Authors described their research in detail, but with minor corrections, in my opinion, the article will be more readable for other researchers.

Abstract:

The abstract should definitely be shortened. Authors should focus on what was the purpose of the work and what the main results were obtained.

Introduction:

The Authors explained why they chose to research, but this is too weakly highlighted. Purpose needs to be redrafted.

The statement: "These functional properties were chosen due to their great interest to the food industry aiming at formulation of wide range of food products" is correct but too obvious.

Results:

The Authors described the obtained results in detail. However, there is no correlation between the methods of electric arc and pulsating electric field influence on the functional properties of beta-lactoglobulin. I suggest the Authors make correlation between the methods used, which will be of interest to other researchers.

Conclusions:

Authors must rewrite this chapter. Authors should only refer to the most important results.

Author Response

Comment 1: “The abstract should definitely be shortened. Authors should focus on what was the purpose of the work and what the main results were obtained.”

Answer 1: Corrected as requested.

Comment 2:The Authors explained why they chose to research, but this is too weakly highlighted. Purpose needs to be redrafted.”

Answer 2: The explanation of the reason of this research is provided in the lines 75-98. The objective of this study has been reformulated as requested (see lines 98-101).

Comment 3: These functional properties were chosen due to their great interest to the food industry aiming at formulation of wide range of food products" is correct but too obvious.

Answer 3: It is true for the readers working in the food industry, but it could be valuable for the readers from other domains. Furthermore, we also mentioned in the manuscript that other functional properties will be studied in perspective (Line 485-487).

Comment 4: The Authors described the obtained results in detail. However, there is no correlation between the methods of electric arc and pulsating electric field influence on the functional properties of beta-lactoglobulin. I suggest the Authors make correlation between the methods used, which will be of interest to other researchers.

Answer 4: Dear reviewer, for each studied functional property, we have provided a statistical difference analysis and discussed the differences between native, PEF-, arc- and heat-pretreated samples. Moreover, we have provided the discussion regarding the duration of PEF and ARC pretreatments. We believe that it should be satisfactory for the readers (see for example L.228-238, 287-290, 296-300, 374-379, etc.).

Comment 5: Authors must rewrite this chapter. Authors should only refer to the most important results.

Answer 5: Corrected as requested also by reviewer 4

Reviewer 4 Report

Dear Authors

 The article is interesting and has scientific potential, and represents a very useful contribution to increase of knowledge in this field, however, it needs some improvement as I will describe in detail below. It is written in very good English. In my opinion, the paper when corrected, the article could be published in the Foods. I recommend MAJOR REVISION.

Abstract

Lines 14-18 – sentences out of place. They rather suit to introduction part. Please rewrite a summary according to MDPI standards.

Lines 31-34 you quote publications on foaming properties that are a little outdated. please quote something more recent. For example:

Coskun and Ocak (2021). Foaming behavior of colloidal whey protein isolate micro-particle dispersions. COLLOIDS AND SURFACES A-PHYSICOCHEMICAL AND ENGINEERING ASPECTS,609, 125660

Nastaj and Sołowiej (2020). Effect of various pH values on foaming properties of whey protein preparations. Int. J. Dairy Technol. 73, 683–694.

Besides - I think it would be worth quoting this article: Šalaševičius et al. (2021) Effect of Pulsed Electric Field (PEF) on bacterial viability and whey protein in the processing of raw milk.

Introduction

please provide data on whether this technique for modifying functional properties of b-lg is safe and can be used in the food industry.

Materials and methods

-how was the protein concentration determined? Did you test it yourself? Kjeldahl method? Manufacturer's data?

why the 20 % b-lg concentration was chosen? Please explain.

-how the b-lg solution was prepared? based on the pure protein content in the preparation?

-the b-lg solutions were stored In the freezer.  Temperature, time and freezer type?

-please harmonise the information in brackets on the origin of the preparations and equipment (company, city, country) according to mdpi standards.

-please provide details about the generator, commercial or self-constructed apparatus?

-You use the term native b-lg? what is the thermal history of b-lg? by writing “native” you are stating that the protein was not heated during b-lg. production. i sincerely doubt it, as it comes from whey and the way comes from the milk, that had to pasteurised for cheese production. Please clarify.

-there are far too many references to other works in the methodology, which makes it difficult to read and know what it is about. Where you can ,please describe it clearly.

-I think that in subsequent studies it would be the best to use rheometer to compare foamabilities of the solutions in question. The surface tension analyses would do the trick too.  it is a pity that the authors did not use these research methods, I think the results would be more interesting.

 Conclusions

Please write your conclusions in prose, without scoring.

please write it in a concise way, outlining and highlighting the most important observations from your research without repeating data you give in your description of results.

please also mention where you see potential industrial use for this modification of functional properties of b-lg.

 Good luck with the corrections!

Author Response

Comment 1: Lines 14-18 – sentences out of place. They rather suit to introduction part. Please rewrite a summary according to MDPI standards.

Answer 1: Corrected as requested

Comment 2: Lines 31-34 you quote publications on foaming properties that are a little outdated. please quote something more recent. For example:

Coskun and Ocak (2021). Foaming behavior of colloidal whey protein isolate micro-particle dispersions. COLLOIDS AND SURFACES A-PHYSICOCHEMICAL AND ENGINEERING ASPECTS,609, 125660

Nastaj and Sołowiej (2020). Effect of various pH values on foaming properties of whey protein preparations. Int. J. Dairy Technol. 73, 683–694.

Besides - I think it would be worth quoting this article: Šalaševičius et al. (2021) Effect of Pulsed Electric Field (PEF) on bacterial viability and whey protein in the processing of raw milk.

Answer 2: dear reviewer, thank you for providing us these three interesting and recently published articles. We have quoted them in the introduction and we will also refer to them for our future studies.

Comment 3: please provide data on whether this technique for modifying functional properties of b-lg is safe and can be used in the food industry.

Answer 3: Dear reviewer, thank you for raising this important point. The PEF is successfully applied in the industrial scale for diverse food treatments while ARC mode is still explored in the laboratory and pilot scales. The phrase was added as requested (L. 73-75).

Comment 4: how was the protein concentration determined? Did you test it yourself? Kjeldahl method? Manufacturer's data?

Answer 4: the protein purity was obtained from the manufacturer (Davisco) (see L.105).

Comment 5: why the 20 % b-lg concentration was chosen? Please explain.

-how the b-lg solution was prepared? based on the pure protein content in the preparation? the b-lg solutions were stored In the freezer. Temperature, time and freezer type?

Answer 5: the requested information was added (L.133-135).

Comment 6: “You use the term native b-lg? what is the thermal history of b-lg? by writing “native” you are stating that the protein was not heated during b-lg. production. i sincerely doubt it, as it comes from whey and the way comes from the milk, that had to pasteurised for cheese production. Please clarify.”

Answer 6: The terminology “native” means the protein non-pretreated protein within the framework of this study, thus representing the negative control. Heat treatment being the positive control. We have provided this additional explanation in the treatment protocol (see line 143-149).

Regarding the second part of your question, the information regarding the thermal history of provided b-lg is confidential and not available. However, from our previous study, where the b-lg structure was explored, we can clearly see the differences in the percentages of secondary structure elements between the native, HVET- and heat-pretreated samples (see ref.26).

Comment 7: I think that in subsequent studies it would be the best to use rheometer to compare foamabilities of the solutions in question. The surface tension analyses would do the trick too.  it is a pity that the authors did not use these research methods, I think the results would be more interesting.”

Answer 7: Thank you for your suggestions. This study is exploratory and we did not have a suitable equipment to perform suggested analyses. However, we will take into account your suggestions in our future studies.

Comment 8: “Please write your conclusions in prose, without scoring.

please write it in a concise way, outlining and highlighting the most important observations from your research without repeating data you give in your description of results.

please also mention where you see potential industrial use for this modification of functional properties of b-lg.”

Answer 8: the conclusion was shortened as requested

Round 2

Reviewer 4 Report

Dear Authors,

The authors have comprehensively addressed all my comments. I have nothing to add, except to recommend this article for publication in the  Foods Journal. Thank you!